# Congruency of Genetic Predisposition to Lactase Persistence and Lactose Breath Test

**DOI:** 10.3390/nu11061383

**Published:** 2019-06-20

**Authors:** Enza Coluccia, Patrizia Iardino, Diego Pappalardo, Anna Lisa Brigida, Vincenzo Formicola, Bruna De Felice, Claudia Guerra, Alessia Pucciarelli, Maria Rosaria Amato, Gabriele Riegler, Laura De Magistris

**Affiliations:** 1Department Precision Medicine, University of Campania Luigi Vanvitelli, 81100 Caserta, Italy; coluccia.enza@gmail.com (E.C.); diego.pappalardo@unicampania.it (D.P.); brigida.annalisa@gmail.com (A.L.B.); claudia.guerra90@libero.it (C.G.); alessia20-07@hotmail.it (A.P.); gabriele.riegler@unicampania.it (G.R.); 2UOC Clinic and Molecular Pathology, University of Campania Luigi Vanvitelli, 81100 Caserta, Italy; patrizia.iardino@unicampania.it; 3Medical Diagnostics Consultant, 80053 Naples, Italy; formicola.vincenzo@fastwebnet.it; 4Department of Environmental, Biological and Pharmaceutical Sciences and Technologies, University of Campania Luigi Vanvitelli, 81100 Caserta, Italy; bruna.defelice@unicampania.it; 5DAI di Medicina Interna e Specialistica, AOU, University of Campania Luigi Vanvitelli, 81100 Caserta, Italy; mariarosaria.amato@unicampania.it

**Keywords:** lactose, lactose intolerance, lactase, H_2_/CH_4_ lactose breath test, lactase genetic test, gastrointestinal symptoms

## Abstract

The physiological decline of lactase production in adulthood, in some individuals, is responsible for the so-called “Lactose Intolerance.” This clinical syndrome presents with gastrointestinal and non-gastrointestinal symptoms following the consumption of dairy containing food. Lactose intolerance can be evaluated by means of the Lactose Breath Test (phenotype) and/or genetic evaluation of lactase-gene polymorphism (genotype). A comparison of the two tests was carried out in a large number of symptomatic adult subjects, which are selected and not representative of the general population. Congruency was as high as 88.6%. Among lactase non-persistent (genotype C/C), 14 subjects showed a negative Lactose Breath Test (LBT), possibly due to young age. Among lactase-persistent (genotype C/T), four subjects showed a positive LBT, which helps to diagnose secondary lactose intolerance. Symptoms, both gastrointestinal and extra-gastrointestinal, were reported by 90% of patients during the breath test. Clinical use of both tests in the same patients could be taken into consideration as a sharp diagnostic tool. We suggest considering the use of the genetic test after LBT administration, when secondary hypolactasia is suspected, for completion of diagnostic procedures.

## 1. Introduction

Lactose is essential for the nourishment of mammalian newborns. The absorption of this disaccharide depends on the activity of lactase, a brush border enzyme cleaving lactose into its monosaccharide components (glucose and galactose) to be directly absorbed. High concentrations of lactase are normally present in the small bowel of all infants that successfully digest lactose provided by human milk or by infant formulas. However, after weaning, there is a decline in intestinal lactase synthesis resulting in low lactase activity in most children worldwide [1]. In mankind, this decline is genetically determined through the down-regulation of the gene LCT (on the short arm of chromosome 2q.21–22). A single nucleotide polymorphism of this gene was described [2], which consists of the nucleotide switch of T for C, resulting in variants CC, CT, or TT-13910. With CC as the original genetic condition, a number of individuals, which varies greatly worldwide, show the persistent genotype (C/T or T/T) that enables them to synthesize sufficient amounts of lactase during a lifespan. This means that adult-type hypolactasia is caused by a cis-acting transcriptional silencing of the lactase gene, and that the individual lactase alleles are regulated independently [1].

However, from a functional point of view, the presence of 50% of the possible lactase activity is enough for adequate lactose digestion [3]. 

Lactose malabsorption can lead to a clinical syndrome in which gastrointestinal symptoms occur, i.e. lactose intolerance. When a sizable fraction of undigested lactose, passing through the small bowel, is delivered to the colon, the osmotic effect will “recall water” influencing motility, and the bacterial fermentation yields multiple products including short chain fatty acids and gases including hydrogen (H_2_), carbon dioxide (CO_2_), and methane (CH_4_). These effects can lead to clinical symptoms: abdominal pain, diarrhea, nausea, flatulence, and/or bloating [4]. Because of the rapid passage of the carbohydrates through the gastrointestinal tract, the symptoms often begin as early as 30 min after ingestion and they can persist up to 6–9 h after food intake [5]. 

The gaseous production of fermentation in the colon is the basis for breath testing. The most diffused, easy, and innocuous diagnostic tool to determine lactose malabsorption is the H_2_/CH_4_ LBT (hydrogen/methane lactose breath test): measurements of breath H_2_/CH_4_ concentrations after ingestion of a standard dose of lactose in which increased gaseous production gives a recognizable breath signal that only occurs in the presence of bacterial fermentation of undigested lactose in the colon [6]. 

Lactase deficiency, however, may be secondary to acquired conditions including: severe malnutrition, mucosal damage due to celiac disease and inflammatory bowel diseases (IBD), bacterial or viral enteritis (e.g., rotavirus), and parasitic disease (e.g., giardiasis, cryptosporidiosis), actinic enteritis, drugs (kanamycin, neomycin, polymycin, tetracycline, colchicine, and other chemotherapeutic drugs), gastrointestinal surgery, short bowel syndrome, and small bowel bacterial overgrowth (SIBO). All these conditions may lead to either a reduction of absorptive capacity or down-regulation of lactase expression in the small intestine [7,8,9,10]. 

The contemporary administration of LBT and genetic testing can contribute toward recognizing a secondary lactase deficiency [3].

The aim of this study was to compare LBT and genetic testing to evaluate their relative reliability and usefulness of contemporary administration. Individual LBT results (phenotype) were compared to the genetic predisposition (genotype) in a large cohort of adult symptomatic subjects, which collects anamnestic and dietary information, as a tool to verify diagnostic results. 

## 2. Materials and Methods

### 2.1. Patients

Over an 18-month period, 158 patients were recruited from among those coming to the Gastroenterology Unit of Azienda Policlinico Università della Campania Luigi Vanvitelli, where the gastroenterologist or family doctor requested they undergo a Lactose Breath Test (LBT) for suspected lactose intolerance.

Besides the prescribed LBT, the enrolled patients accepted to undergo genetic testing, for which peripheral whole blood (4 mL) was collected. All subjects gave their informed consent for inclusion before they participated in the study. The study was conducted in accordance with the Declaration of Helsinki, and the protocol was approved by the Ethics Committee of Università della Campania Luigi Vanvitelli (Project identification number 685, 18-12-2017).

Exclusion criteria were: age <18 and >75 years, major pathological and/or metabolic diseases, and pregnancy.

There were 101 females and 57 males, with age ranging from 18 to 74 years (mean ± SD = 40.4 ± 14.9; median 38 years), with all of them being Caucasian and coming/living from South Italy.

An anamnestic/morphometric questionnaire was administered to all patients. They were also asked about alimentary habits and the presence of gastrointestinal (GSRS scale) [11] and extra-intestinal symptoms (extra-GI scale) (adapted from Reference [12]).

### 2.2. Lactose Breath Test (LBT)

In preparation, all participants were prevented from ingesting fiber-rich food on the day before the test. They were asked to consume a light dinner with rice and meat/fish with no fibers. Antibiotic or laxative therapies, as well as pre/probiotics during the previous two weeks were not allowed. Smoking and physical exercise were not permitted during the test. 

After an overnight (at least 10 h) fast, a first sample of breath was obtained to evaluate basal gaseous excretion of H_2_ and CH_4_. H_2_ basal values > 30 ppm and CH_4_ > 5 ppm were considered exclusion criteria for performing the test. Lactose—25 gr suspended in 250 mL water—was then administered. Breath samples were obtained every 30 min for 4 h. Each breath sample was examined by means of a Mycrolizer DP plus H_2_/CH_4_ analyzer (Quintron, USA). The test was considered positive if any of the H_2_ plotted values was higher than 20 ppm over the basal value and/or CH_4_>12 ppm.

In the presence of a two-peak pattern, patients were invited to return to be administered a standard glucose breath test (GBT, glucose 25 g oral load) to exclude Small Intestine Bacterial Overgrowth presence (SIBO). In the presence of SIBO the patients were excluded [13].

When a longer orocecal transit time was suspected, additional breath samples were taken up to 5 h.

### 2.3. Genetic Test

Genomic DNA was isolated from EDTA-uncoagulated blood by using Eu-Gen extraction kit (Eurospital SpA, Trieste, Italy), where DNA concentration of each sample must be between 10 and 100 µg/mL, followed by R.T. PCR amplification, specific for T/T, C/C, and C/T genotype (LactoGen kit-Eurospital SpA, Trieste, Italy). The analytical specificity of LactoGen kit had been evaluated by aligning and comparing the sequences of the trigger oligonucleotides and the fluorescent probes with the sequences available at the nucleotide databases [14,15]. 

## 3. Results

1. LBT testing resulted positive in 75.9% (*N* = 120) and negative in 24.1% (*N* = 38) subjects.

2. The genetic test resulted in 82.3% C/C (*N* = 130), 17.7% C/T (*N* = 28) and no T/T subjects.

3. When comparing the individual LBT result and genetic predisposition, we found 88.6% congruency. Four lactase-persistent (C/T) subjects (14.8% of 28) showed a positive LBT (*p* < 0.01 Chi squared test) and 13 non-persistent (C/C) subjects (10.5% of 130) showed a negative LBT (*p* < 0.01 Chi squared test). 

Detailed data of the two groups of subjects are shown in Table 1 and Table 2.

4. Among LBT positive patients, 90% developed symptoms (Figure 1). One only among LBT negative values (migraine). Most frequently reported were: abdominal distension 83.3% (*N* = 100), gastric/esophageal burning 41.7% (*N* = 50), abdominal pain 25% (*N* = 30), diarrhea 10% (*N* = 12), and headache/migraine 27.5% (*N* = 33).

## 4. Discussion

The reported amount of lactose intolerance in the Italian general population, when investigated by means of LBT, is 50% [16]. In Southern Italy, a prevalence of 41% was reported [17]. The obtained percent of LBT positive subjects among the study sample in the present study is higher (i.e., 76%) than the above reported amounts. The presently investigated sample is not, however, representative of the general population. The investigated subjects were chosen among those already suspected to be intolerant and, hence, addressed to undergo to LBT by the family doctor. 

The lactase persistent phenotype is highly variable worldwide. In Europe, a North-South gradient can be found. In Scandinavian countries, 90% of the adult population can digest lactose, while, in Mediterranean populations, tolerant adults are less than 50% [18,19,20]. In Italy [21,22], a global frequency of 62.3% non-persistent genotype (C/C) was recently reported [23]. In Southern Italy, the frequency resulted in 67.1%. Again, in the present study, the higher percentage of C/C subjects (82.9%) could be due to the selected sample that is not representative of the general population.

To bear the C/C genetic condition means that lactase expression during a life span will almost certainly and consistently diminish. Age is the first determining factor. In most subjects, this starts to happen early in life.

We have no instruments, however, to foresee when this phenomenon will start and not even its individual extent. A total lack of lactase is rare and its diminution is subjective [10]. Among the presently investigated subjects, 120 resulted C/C, and, among them, 13 showed a negative LBT. Among them, 11 out of 13 C/C showing a negative LBT were younger than the median age of the sample (38 years). Therefore, this cannot be considered incongruent in comparing the two diagnostic tests. The genetic test alone, however, is not sufficient to diagnose a lactase deficit in any non-persistent subject. A second factor influencing the capacity to digest lactose is the possible domestication of lactic acid bacteria, which ferment the not-digestible lactose to easily absorbed lactic acid [24]. Although lactase expression is not up regulated by lactose ingestion, it has been reported that regular intake of even small amounts of lactose may improve tolerance through the adaptation of the intestinal microbiota [25]. Ten of the 13 C/C subjects showing a negative LBT, regularly consumed dairy products and occasionally yogurt. Three out of 13, instead, had already withdrawn any dairy containing food from their diet since at least 6 months. The present data only partially support the hypothesis of intestinal adaptation based on the presence of lactose-digesting bacteria. It is well known that rare false negative results are described during breath testing due to limitations of conventional LBT. It can depend on improper test preparation or execution [3]. We exclude this event because only patients appropriately asked and controlled upon these aspects were enrolled. Besides, the LBT-negative did not report any symptom during the test, but one with a migraine. False negative can be due to hydrogen non-excretion [26] and/or to longer orocecal transit time because the test may be finished before a measurable H_2_/CH_4_ increase is established [27]. The combined measurement of H_2_ and CH_4_ excretion allowed us to recognize ‘‘low H_2_ producers’’ [28] such as one of the C/T patients reported in Table 1 (case N°26). The possible presence of a longer orocecal transit time was overwhelmed by additional breath samples that take up to 5 h.

Therefore, it is not possible to detect a Secondary Lactase Deficit (SLD) in any of the non-persistent subjects with a positive LBT. In the present study, 107 non-persistent lactase subjects (C/C) had a positive LBT. They were recruited because they were unaffected by any main intestinal pathology, such as celiac disease or IBD, but it is unknown if they were affected by Secondary Lactase Deficit (SLD).

To bear the C/T or T/T genetic condition means that there will be no lowering of lactase expression during life, or that there will be a lowering of its activity, which allows it to digest lactose amounts normally present in the diet. It also means that, in such subjects, the LBT, at the experimental conditions, should be negative. In the present study, four lactase-persistent subjects showed positive LBT. It represents 6.3% real incongruence between the two diagnostic tests. As a matter of fact, a Secondary Lactase Deficit (SLD) can be diagnosed, in the lactase-persistent subjects, only by the administration of both the LBT and the genetic test, as already suggested [29]. A false positive LBT test can be due to rapid GI transit/SIBO. Peak time, however, allowed us to further investigate suspected cases and none of the enrolled patients were affected by SIBO. In the present study, four subjects were found to have an SLD. In all of them, the presence of SIBO was excluded. In one of them, undiagnosed Celiac Disease was suspected. 

Originally, lactase deficiency was defined by the measurement of enzymatic activity (lactase versus sucrase activity ratio) in duodenal/jejunal biopsies [30] or through a more recently described biopsy-based test known as the Quick Test [31]. Since their description and evaluation [32,33], the less invasive, lactose-tolerance tests are now used, such as the measurement of breath hydrogen increase after a lactose load (Lactose breath test, LBT) that have shown sensitivity of 76% to 100% and specificity of 90% to 100% [6,34]. Even though it is known that a standard (25 g) lactose breath test for adults may lead to underestimation of lactose malabsorption [35], in the routine management, we use this dosage following previous discussed criteria [36].

Several studies have recently been conducted to study the correlation between genetic and breath testing to diagnose lactase deficiency. They generally find very good agreement between the two tests [29,37,38,39,40,41] or only moderate [27]. We also found a very high concordance between the two diagnostic methods. However, this study underlines substantial differences among them and the unique ability to diagnose secondary lactase deficiency (SLD) through the contemporary administration of both tests.

As already reported [3], patients developed gastrointestinal symptoms during LBT, with abdominal distension being the most common. We found that gastric burning was rather commonly reported (41.7%) even though we did not find any report of it in the specific literature. The rather frequently reported headache/migraine, even in one of the LBT negative patients, could be explained by factors other than lactose ingestion, such as prolonged fasting.

## 5. Conclusions

We suggest considering the use of genetic tests after LBT for people belonging to Southern Italy as completion of diagnostic procedures, when secondary hypolactasia is suspected.

## Figures and Tables

**Figure 1 nutrients-11-01383-f001:**
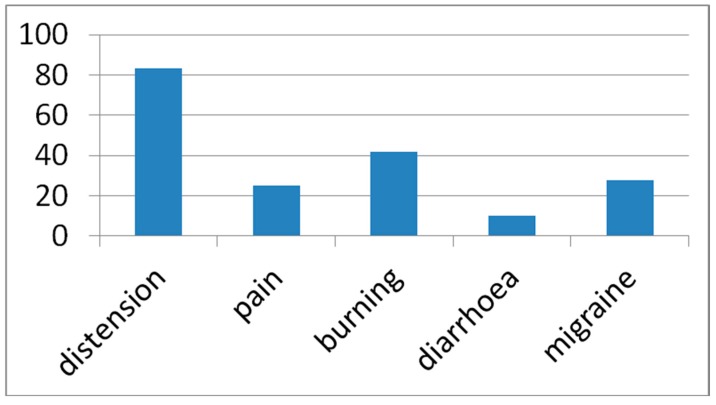
Lactose related symptoms were reported by 90% of LBT positive patients. Data are expressed as the percentage of the total sample. Distension = abdominal distension. Pain = abdominal pain. Burning = hearth (gastro/esophageal) burn.

**Table 1 nutrients-11-01383-t001:** Anagraphic, morphologic, and breath test details of the four C/T subjects in whom LBT and genetics are not congruent. Case N°26 is methane producer and H_2_ negative.

N°/Name	Sex/Age	BMI	Genetics	Basal Breath H_2_/CH_4_ (ppm)	Peak Value ∆ppm	Peak Time
26 AV	F 51	21.4	C/T	4 CH_4_ 2 H_2_	29 CH_4_ 4 H_2_	180
46 NAN	F 18	19.5	C/T	5	77	120
59 PR	F 69	22.9	C/T	15	153	90
157 SB	F 26	19.6	C/T	30	76	120

**Table 2 nutrients-11-01383-t002:** Anagraphic, morphologic, and breath test details of the 13 C/C subjects in whom LBT and genetics are not congruent.

N°/Name	Sex/Age	BMI	Genetics	Basal Breath H_2_/CH_4_ (ppm)	Max Registered Value
2 PC	M 31	20.0	C/C	3	4
20 MF	M 28	33.0	C/C	6	6
28 VL	F 42	27.0	C/C	15	20
32 MS	F 22	32.5	C/C	5	14
38 CA	F 19	29.0	C/C	12	17
75 VL	M 20	23.1	C/C	2	2
76 DM	F 24	16.6	C/C	3	2
82 DCA	M 22	26.7	C/C	5	22
104 AC	M 57	28.4	C/C	8	8
124 PS	M 72	29.7	C/C	12	15
134 NA	M 29	24.8	C/C	2	11
142 SV	M 23	28.3	C/C	14	18
149 RG	F 33	36.3	C/C	1	2
151 DNL	F 38	41.1	C/C	4	8

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
