# Peer review of "Congruency of Genetic Predisposition to Lactase Persistence and Lactose Breath Test"

_nutrients, 2019, doi:10.3390/nu11061383_

Round 1

Reviewer 1 Report

This is a very interesting study that aimed to determine concordance of the Lactose Breath Test (LBT) and genetic testing in determining lactose intolerance, versus other conditions that may lead to secondary lactase deficiency. The study found over 80% agreement between the two diagnostics methods.

The manuscript is clear and well written. However, other than a brief mention regarding recruiting lactic acid bacteria as members of the gut microbiota (note: the term “microflora” is not currently favored) to help digest lactose, there is no discussion about the role of lactose-digesting gut bacteria in the discordant individuals. For example, do these individuals consume more lactose-free dairy products like yogurt? If so, relative abundance of lactose-digesting bacteria may be significantly higher.

Although the text is clear, the tables are difficult to interpret. A graph (or graphs) would be more visual and easier to read. Finally, a more thorough statistical analysis would add to this publication.

Author Response

The manuscript is clear and well written. However, other than a brief mention regarding recruiting lactic acid bacteria as members of the gut microbiota (note: the term "microflora" is not currently favored) to help digest lactose, there is no discussion about the role of lactose-digesting gut bacteria in the discordant individuals. For example, do these individuals consume more lactose-free dairy products like yogurt? If so, relative abundance of lactose-digesting bacteria may be significantly higher.

ANSWER: The term microflora was changed to microbiota. For each patient, a short questionnaire on alimentary habits was collected. Among the 13 C/C LBT negative subjects, ten regularly assumed dairy products while 3 were already on lactose free diet. None used to eat yogurt every day. This aspect is now discussed, see lines 169-173 on page 5.

Although the text is clear, the tables are difficult to interpret. A graph (or graphs) would be more visual and easier to read.

ANSWER: Very sorry for the tables; in our opinion there is no better way to show those data!! We tried to ameliorate them. A figure was added to describe reported GI symptoms.

Finally, a more through statistical analysis would add to this publication.

ANSWER: Chi-squared test has been applied to data and results are shown on page 3, lines 125-127.

Reviewer 2 Report

I have reviewed the study by Coluccia et al submitted to Nutrients. The study consists of a cohort of southern Italian adults prospectively evaluated for lactose intolerance. The aim of the study was to assess congruency between the lactose breath hydrogen test (25g lactose challenge) and the genetic test (DNA, PCR) for C/T-13910 Caucasian European polymorphism. In comparing the breath test 75.9% were positive and 24.1% were negative among 158 recruited Caucasian patients. Using the genetic tests 82.3 % were wild type C/C (lactase non persistent, LNP) and 17.7% were heterozygote C/T (phenotypic lactase persistent, LP). Exclusion criteria were appropriate, Data on pretest diet, gastrointestinal symptoms and extra intestinal; symptoms were tabulated. Congruency between LBT and Genetic tests 88.6%. Of the lactase non –persistent group 13 were LBT negative while 4 LP patients had a positive LBT suggesting secondary lactase deficiency in these participants.

Comments:

 I think the study is well done and of interest although not highly novel anymore. What I find interesting actually is the 13 patients who were LNP but had negative gas production on LBT testing. The authors mention several possibilities for this outcome. Line 146 - 153 starts the explanation with age, and then colonic adaptation. On pg 5 of the discussion line 180 then discusses further reasons for their findings with false positive negative. I think these 2 paragraphs would be better to be connected in the discussion.

As a further comment the authors state they have gathered data on dietary intake. Were any of the LNP, LBHT negative patients consuming regular dairy which could be an example of “deep” adaptation where then withholding dairy foods for a few weeks and retesting might prove positive LBT on the second test?

On pg 4 line 129 I think the tabulation of gastrointestinal and extra-gastrointestinal symptoms could be better represented than a foot note to table on false results in LP or LNP patients. Possibly a separate table or at least a separate paragraph outlining symptoms I think would be better.. Some of the extra intestinal GI symptoms are more novel e.g. as the authors point out; heartburn.

Finally there are many syntax errors using English starting in the Abstract eg. Line 18 not-gastrointestinal symptoms -----should be non –gastrointestinal . Same line assumption of dairy should be---consumption

Line 22- Congruency resulted ---change to was as high etc…. The paper should be reviewed by an English proficient person.

Author Response

I think the study is well done and of interest although not highly novel anymore. What I find interesting actually is the 13 patients who were LNP but had negative gas production LBT testing. The authors mention several possibilities for this outcome. Line 146-153 starts the explanation with age, and then colonic adaptation. On pg 5 of the discussion line 180 then discusses further reasons for their finding with false positive negative. I think these two paragraphs would be better to be connected in the discussion. 

ANSWER: As suggested, the paragraphs were connected and discussion unified; see lines 169-182 on page 5. 

As a further comment the authors state they have gathered data on dietary intake. Were any of the LNP, LBHT negative patients consuming regular dairy which could be an example of "deep" adaptation where then withholding dairy foods for a few weeks and retesting might prove positive LBT on the second test?

ANSWER: For each patients a short questionnaire on alimentary habits was collected. Among the 13 C/C LBT negative patients, 10 regularly assumed dairy products while 3 were already on lactose free diet. This aspect is now discussed, see lines 169-173 on page 5. We did not retest any patient, thank you for the suggestion.

On pg4 line 129 I think the tabulation of GI and extra-GI symptoms could be better represented than a foot note to table on false results in LP or LNP patients. Possibly a separate table or at least a separate paragraph outlining symptoms I think would be better. Some of the extra-intestinal GI symptoms are more novel e.g. as the author point out: heartburn.

ANSWER: GI and non-GI symptoms are now described in paragraph 4 of the Results; a figure has also been added.

Finally there are many syntax errors using English starting in the abstract e.g....... etc. The paper should be reviewed by an English proficient person.

ANSWER: the suggested mistakes were fixed and a through revision of language was done by an English proficient expert person